# Connexin-43 in Cancer: Above and Beyond Gap Junctions!

**DOI:** 10.3390/cancers16244191

**Published:** 2024-12-16

**Authors:** Shishir Paunikar, Luca Tamagnone

**Affiliations:** 1School of Medicine, Università Cattolica del Sacro Cuore, 00168 Rome, Italy; shishir.paunikar@unicatt.it; 2Fondazione Policlinico Universitario “A.Gemelli” IRCCS, 00168 Rome, Italy

**Keywords:** connexin-43, cancer, progression, non-canonical, c-terminus, hemichannels

## Abstract

Connexin-43 (Cx43) is known to be involved in Gap Junctional Intercellular Communication (GJIC). However, Cx43’s non-canonical functions, encompassing signaling pathways elicited by its C-terminal tail and hemichannel activity, have gained traction in cancer biology. This review article focuses on the pro-tumorigenic implications of Cx43’s non-GJIC functions and explores Cx43-dependent mechanisms contributing to cancer progression.

## 1. Connexin Proteins

Connexins are four-pass transmembrane proteins, responsible for the formation of inter-cellular communicating gap junctions (GJs), as well as for the assembly of hemichannels (HCs) that allow molecule exchange between the cytosol and the extracellular environment. In both settings, connexins act as pivotal players of signaling cascades. In fact, these channels can vehicle ions, metabolites, secondary signaling molecules (up to 1.5 kDa), and small noncoding RNAs (siRNA and miRNA) [1,2]. A hemichannel, also known as a connexon, is formed by the assembly of six connexin (Cx) subunits embedded in the cell plasma membrane. The joining of hemichannels between two adjacent cells leads to the formation of a complete gap junctional channel, directly connecting the cytoplasm of the neighbor cells. Each connexin subunit contains two extracellular loops and one cytoplasmic loop, while both the N-terminal and C-terminal regions are located within the cell [3]. Hemichannels may also be formed by pannexin proteins, which share the same topology but lack sequence homology with connexins [4].

With very few exceptions, connexin genes share a common structure, comprising two exons separated by a single intronic sequence [5]. Evolutionary analysis of connexin sequences indicates their specificity to chordates, with a high degree of genetic conservation. Diverse connexin isoforms are expressed differentially, exhibiting spatial and temporal specificity, under the control of transcription factors [6]. Each connexin is typically identified based on its predicted molecular weight; for instance, connexin-32 (Cx32) has an approximate molecular weight of 32 kDa; moreover, for non-human counterparts, a prefix is added to notify the species specificity [3,7]. The connexin gene nomenclature has been further standardized, with the prefix “GJ” used to denote gap junctions, and four classes (A, B, C, and E) based on sequence similarities. Within each class, individual genes are assigned an Arabic numeral, which largely reflects the order of their discovery (e.g., Cx32 is designated as GJB1) [3]. There are 21 distinct human connexin genes which encode proteins sharing a similar structural topology. While the connexin family exhibits a high degree of genetic conservation, the primary variation among members lies in the length and sequence of the cytosolic C-terminus (CT), also known as the carboxyl or C-terminal tail [8]. Notably, this C-terminal region appears to have a crucial role in a range of activities, encompassing the gating of gap junction channels and hemichannels, as well as non-canonical connexin functions beyond their channel-forming capacity [9,10,11,12,13].

## 2. Connexin-43 (Cx43)

### 2.1. Structure of Cx43

The most abundantly expressed, evolutionarily conserved, and best studied connexin is connexin-43 (Cx43). Cx43 forms hexameric structures called connexons, which dock with connexons from adjacent cells to form gap junction channels. These channels allow for the selective passage of molecules, depending on their size and charge. Cx43 channels can also exist as hemichannels that open to the extracellular environment, playing roles in paracrine signaling and modulating the extracellular environment.

The Cx43 carboxyl terminus (CT) consists of approximately 150 amino acids, representing about 39% of the total protein length (Figure 1). This segment is notably enriched in prolines and serines, which bears significant implications and designates Cx43-CT as a prospective site for protein interactions (with cytoskeletal molecules and potential oncogenes) and post-translational modifications [13].

### 2.2. Functions of Cx43

Cx43 is predominantly studied in its capacity to form gap junctions, intercellular channels that allow for direct communication between adjacent cells; this phenomenon, referred to as Gap Junctional Intercellular Communication (GJIC), is crucial in various tissues, particularly in the heart, nervous system, and bone. Moreover, increasing evidence points to Cx43 involvement in cancer [12,14].

In light of its GJIC activity, Cx43 maintains tissue homeostasis in physiological processes, such as development, wound healing, and immune responses [15,16,17]. In fact, gap junctions formed by Cx43 enable the exchange of ions, secondary messengers, and metabolites, which help to coordinate cellular activities [18,19]. Cx43 is further involved in neuronal protection [20], as it is expressed in astrocytes, where it helps to maintain the blood–brain barrier and to modulate inflammatory responses after brain injuries [21,22]. In the heart, Cx43 plays a critical role by enabling the coordinated contraction of cardiac muscle cells [23,24]. It facilitates the passage of ions and small molecules between cardiomyocytes, which is essential for the propagation of electrical impulses across the heart [25,26,27]. Consequently, Cx43 mutations resulting in functional dysregulation have been linked to arrhythmias and other cardiac diseases [28,29]. Cx43 is also involved in regulating osteocytic communication in bone tissue [30], helping to maintain bone density and integrity by facilitating the transmission of signals for bone remodeling [31,32]. Indeed, skeletal abnormalities have been associated with Cx43 mutations and dysregulations [33].

On the other hand, the role of Cx43 in cancer appears complex and context-dependent. It was initially believed that Cx43 acts as a tumor suppressor by coordinating normal cell communication through its GJIC function, cumulatively preventing uncontrolled cell proliferation. Cx43 tumor suppressor mechanisms involve mediating pro-apoptotic signals through gap junctions, and reducing migration and invasive capacity by maintaining cell–cell adhesion; as well as modulating the tumor microenvironment, by exchanging secondary messenger molecules through Cx43 hemichannels [13,14,34,35,36,37]. Thus, in certain cancers, Cx43 downregulation may lead to disrupted cell communication and contribute to tumor progression [38]. However, the accumulating literature has reported instead a tumor-promoting role of Cx43, especially in advanced cancers, which may be accounted by its non-canonical, gap junction-independent, functional activities. In this review we provide an overview of the role of Cx43 in cancer cells, focusing on its potential involvement in tumor metastatic progression.

## 3. Connexin-43 in Cancer

### 3.1. Cx43 Activities in the Tumor Microenvironment

Recent studies suggested that Cx43 might facilitate cancer cell survival by mediating anti-apoptotic and growth signals via gap junctions, as well as promote metastasis by modulating cancer cell motility and interactions with ECM components [14,39,40,41,42]. It is also suggested that Cx43 can control angiogenesis and immune cell evasion in the tumor microenvironment [43,44,45,46,47]. A pro-proliferative role of Cx43 was assessed in Multiple Myeloma (MM) cells, where GJIC activity between Bone Marrow Stem Cells (BMSCs) and MM cells has been evidenced [48]. Similarly, Cx43 gap junction function was shown to be involved in inhibiting pro-apoptotic signals in cancer, where blocking Cx43 GJIC activity with chlordane or RNAi induced the apoptosis of hepatoma cells [49]. In the same vein, Jensen et al. showed that a carbenoxolone (CBX) blockade of Cx43 sensitizes thyroid cancer cells to anoikis [50]. Furthermore, the pro-metastatic involvement of Cx43 GJIC is vehemently described in various cancers including prostate [51], lung [52], breast [53], melanoma [54], glioblastoma [55] and brain [56].

Notably, recent research unveiled a multifaceted functional role of Cx43 in cancer cells, beyond its known involvement in junctional function (Figure 2). For instance, Cx43 has been implicated in the regulation of intracellular pathways modulating cell growth and cell death, in mechanical signal transduction, and in the regulation of gene transcription [57,58]. These aspects will be addressed in detail in Section 4. For example, the C-terminal tail of Cx43 (Cx43-CT) can interact with various oncogenes and may serve as a trigger for intracellular signaling cascades, control metabolic pathways, impact transcriptional control, and potentially be released within extracellular vesicles featuring paracrine cell–cell communication in the tumor microenvironment through its hemichannel function [11,59,60,61]. Other studies have highlighted that Cx43 can facilitate communication between non-adjacent cells, through specific structures like tunneling nanotubes (TNTs), as well as extracellular vesicles [12,60,62]. Notably, Cx43-based hemichannels are suggested to influence tumor progression. Cx43 hemichannels facilitate the release of cell proliferation/survival signaling molecules like NAD+ (nicotinamide adenine dinucleotide), cADPR (cyclic ADP-ribose), ATP, Ca^2+^, IP3, glutathione, and prostaglandin E2, which can influence cancer cell survival, migration, and invasion by modulating the extracellular environment, including altering ion concentrations and releasing growth factors [63,64,65].

### 3.2. Altered Expression of Cx43 in Human Tumors and Its Functional Impact

Alterations of Cx43 levels, encompassing both over- and under-expression, have been associated with initial and advanced stages of different malignancies [40,66,67,68]. Actually, the current literature presents contrasting findings, with some studies suggesting that Cx43 functions as a tumor suppressor [36,66,69], while an increasing body of evidence points to its pro-tumorigenic role [10,13,70]. For instance, Cx43 has been found to be upregulated in human invasive breast carcinomas compared to both normal mammary epithelial cells and tumor cells from non-invasive lesions [71]. Increased expression of Cx43 has been found in lymph node metastases of breast cancer, in contrast to primary tumors, where its expression appeared to be lower compared to normal mammary epithelium [72]. In another study, Cx43 elevated expression was associated with poor prognosis in estrogen receptor-negative but not other subtypes of breast cancers [73]. Increased Cx43 mRNA levels in gliomas have been correlated with lower patient survival rates [74], and its expression has been associated with increased glioma cell motility and invasiveness [75]. In contrast, Cx43 expression was found to be lower in non-small cell lung cancers compared to adjacent normal tissues, owing to promoter methylation, which prevents *GJA1* gene activation [76]. However, overexpression of Cx43 was also shown to facilitate the adhesiveness of cancer cells to lung endothelial cells, promoting metastatic extravasation [67]. In a study conducted on bile duct carcinoma cells, it was observed that Cx43 acts as a tumor promoter, since its inhibition resulted in a discernible reduction in cancer cell proliferation [77]. A higher expression of Cx43 has been correlated with the upregulation of Snail-1, which is a transcription factor implicated in suppressing E-cadherin and promoting epithelial–mesenchymal transition (EMT); this change resulted in higher motility and invasiveness of prostate cancer cells [78]. Notably, in advanced metastatic colon cancer lesions, point mutations were identified in the multifunctional carboxyl-terminal domain of Cx43 [79], which were putatively associated with cancer invasiveness.

### 3.3. Disrupted Cx43 Localization in Tumors

Irregular or disrupted connexin subcellular localization is a pivotal marker of tumoral development. In particular, nuclear localization of Cx43 has been observed in several tumors, such as colon cancer and gliomas, as well as breast carcinomas [69,80,81]. This is often due to disruption of connexin trafficking [82] or impaired degradation mechanisms [83]. The implications of this intracellular mislocalization await elucidation; however, recent findings have suggested that nuclear Cx43 (or its carboxyl terminus alone) can function as a direct transcription factor [43].

In general, Cx43 expression and mislocalization have been reported to vary, depending on tumor type and stage of cancer progression [84], with a trend of elevated expression in metastatic lesions [85]. The multifaceted involvement of Cx43 in diverse human cancers is summarized in Table 1, below.

### 3.4. Cx43 and Mitochondrial Regulation

Notably, mitochondrial Cx43 has also been involved in the apoptotic process of cardiac myocytes, in a study conducted by Goubaeva and coworkers. This is achieved by boosting the release of Ca^2+^ and cytochrome C from isolated mitochondria, following inactivation of GJIC [9,135]. Furthermore, in pancreatic cancer cells, the interaction between Cx43 and the pro-apoptotic protein Bax contributes to the activation of the mitochondrial apoptotic pathway [128]. This suggests that Cx43 may promote apoptosis in cancer cells in certain instances, leading to mitochondrial dysfunction and the subsequent release of apoptotic factors.

### 3.5. Cx43 Hemichannels in Cancer

Cx43-based hemichannels have been implicated in promoting tumor growth. For example, a recent study by Khalil et al. suggests Cx43 hemichannels induce collective cancer cell invasiveness by releasing ATP into the extracellular environment and initiating an autocrine purinergic signaling loop. In fact, ATP activates P2Y2 receptors on the cell surface, which in turn leads to the recruitment of Src kinase and the ERK signaling pathway, promoting collective cell migration and invasion [87]. It was also observed that blocking Cx43 hemichannels with antibodies directed against the second extracellular loop domain hinders glioma tumor growth in rat models inoculated with C6 glioma cells [136]. Furthermore, the peptide αCT1, which selectively inhibits Cx43 hemichannel activity, has been shown to enhance the sensitivity of glioma cells to chemotherapy, featuring a potential therapeutic strategy to improve the efficacy of chemotherapy in gliomas [74]. These data suggest that Cx43 hemichannels play a role in promoting metastatic properties in tumor cells. This promising new research avenue is gaining traction, and the relationship between Cx43 hemichannels and cancer metastasis will be explored further in Section 4.2 of this review.

### 3.6. Cx43 and Tunneling Nanotubes

Tunneling nanotubes (TNTs) are long and thin membrane projections that connect cells, facilitating the exchange of not only small molecules, but also larger proteins, organelles, bacteria, and viruses. Typically, TNTs exhibit increased formation under conditions of cellular stress and are more prominent in cancer cells, where they are generally thought to be pro-metastatic and to provide growth and survival advantages. Interestingly, Cx43 positioned at the tips of TNTs has the capacity to connect non-adjacent cells [62]. This physical integration plays a pivotal role in the creation of metastatic clusters and contributes to angiogenesis by coupling endothelial cells and pericytes [46]. A recent study conducted by Sinha et al. suggested TNTs, facilitated by Cx43, to be involved in the transfer of the metastasis-related protein GIV to estrogen receptor-positive (ER+) breast cancer cells [137]. *CCDC88A*/GIV is involved in potentiating tumor cell survival, invasion, and chemoresistance [138], and was shown to interact with Cx43 and co-migrate from Mesenchymal Stromal Cells (MSCs) to ER+ breast cancer cells via Cx43–TNT mediated intercellular transfer, ultimately facilitating metastatic aggression [137].

### 3.7. Cx43 in Cancer Stem Cells

Interestingly, Cx43 has been implicated in the regulation of cancer stem cells (CSCs) [139]. For instance, increased expression of Cx43 was observed upon the differentiation of glioblastoma stem cells (GSCs), and certain subtypes of breast cancer stem cells that express the putative CSC marker CD44 also exhibit Cx43 expression [14,140]. It has also been reported that cancer stem cells exploit connexins to support their aggressive behavior. For instance, a study conducted by Patel et al. showed Cx43 to facilitate immune evasion in breast cancer by enabling communication between cancer stem cells and mesenchymal stem cells [141]. Similarly, lung cancer metastasis to the brain is shown to be promoted by Cx43, connecting cancer stem cells and astrocytes [142]. Furthermore, according to Zeng et al., Cx43 mislocalization in the cytoplasm of non-small cell lung carcinoma cells leads to the upregulation of the stemness markers Oct4 and NANOG [82].

Despite consistent evidence of the tumor-promoting role of Cx43, it should be recalled that its GJIC activity has also been associated with tumor-suppressor functions [69,143,144,145,146,147,148]; this is consistent with the general knowledge that loss or disruption of gap junctions and of the associated communication is an early event in tumor transformation [2,149,150,151]. It is therefore worthwhile delving deeper into the non-canonical and gap junction-independent roles of Cx43 in cancers to help understand its role in metastatic progression. These gap junction-independent functions have recently come into the limelight with extensive research being focused on the carboxyl-terminal portion of Cx43 (Cx43-CT), which acts as a hub for protein–protein interactions. The role of Cx43-CT in the context of cell signaling and its implications for tumor biology will be discussed in the subsequent section.

## 4. Mechanisms of Gap Junction-Independent Activity of Connexin-43 in Cancer

### 4.1. Cx43 Carboxyl Terminal Tail Interactions

Gap junction independent functions of Cx43 have garnered attention in recent years owing to their implication in cancer development and progression. Primarily, these interactions are orchestrated by the CT domain, which comprises multiple interacting motifs. Indeed, various studies have shown that Cx43-CT engages in a range of protein–protein interactions that operate autonomously from gap junction channel functionality [11]. For example, the Cx43-CT domain directly interacts with cytoskeletal and scaffolding proteins, including both α- and β-tubulin (through a 35-amino acid juxtamembrane segment), allowing Cx43 to anchor the distal ends of microtubules at gap junctions and influence microtubule dynamics in adjacent cells [152]. Additionally, Cx43-CT binds to the second PDZ domain of ZO-1 [153], a member of the membrane-associated guanylate kinases (MAGUKs), which are multidomain scaffolding proteins involved in cell signaling and structural organization [154].

Cx43 mislocalization is key to its intracellular interactions; indeed, connexin proteins have been found to localize even in the nucleus, where they can initiate the transcription of target genes. Intriguingly, multiple studies showed that Cx43-CT contains a putative nuclear targeting sequence, implicated in both anti- [155,156] and pro-tumoral effects [157,158]. An in-depth investigation was conducted by Crespin et al. to discern the role of Cx43 in modulating the actin cytoskeleton in human glioma cells [80]. This study employed a construct lacking the CT domain, as well as another variant devoid of the transmembrane domain. Indeed, a comparable decline in the proliferation of glioma cells was observed in the presence of either of the constructs. Notably, truncated Cx43 had no discernible impact on gap junction activity, while the Cx43-CT domain alone was competent to stimulate the migration of glioma cells [38].

It is also reported in various studies that this CT domain can intersect mitogenic signaling pathways and the cell cycle regulation apparatus, enabling Cx43 to impinge on cell growth, differentiation, and migration [159]. This is due to interactions with known regulatory molecules, such as CCN family proteins, which are multi-modular proteins involved in cell adhesion, migration, and proliferation [160]. In particular, CCN3 (NOV) interacts with Cx43-CT independent of its gap junction function, inhibiting cell growth [161,162,163]. Also, CCN3 (NOV) may enhance Cx43-mediated cell adhesion by regulating Rac1 and the actin cytoskeleton [164]. Additionally, Cx43-CT interacts with Hsc70 [165], a molecular chaperone, negatively regulating the nuclear translocation of the CDK inhibitor p27, which can explain its impact on cell cycle progression [166]. It was shown that Cx43-CT directly promotes the expression of the mesenchymal marker N-cadherin by directly interacting with the promoter region of the N-cadherin gene [157]. Behrens et al. found that expression of the Cx43-CT domain (amino acids 257–382) increased HeLa cell migration via modulation of p38 MAP kinase activity [167]. A study by Dai et al. reported that Cx43-CT can displace Smad2/3 interaction with microtubules (MTs), thereby enhancing Smad2 phosphorylation, which consequently leads to its nuclear localization and the transcriptional activation of target genes [168]. Furthermore, through its association with the actin-binding proteins vinculin and debrin, Cx43 was implicated in the directional motility of cardiac neural crest cells, independent from its channel-forming capacity [169]. The interaction of Cx43-CT with the skin tumor suppressor caveolin-1 (Cav-1) was studied in rat epidermal keratinocytes by Langois et al., revealing that Cx43 downregulation promotes EMT and fosters invasive traits in skin cells [170]. Cx43 is also reported to directly interact with Integrin-α5 (through Ser-373 in the CT), which promotes the opening of hemichannels [171]. Indeed, the interactions of Cx43 hemichannels constitute a distinct and increasingly recognized area of research, particularly regarding their role in modulating cancer cell behavior. These interactions merit further research and detailing, as they contribute significantly to cancer progression and highlight an additional viewpoint of Cx43’s multifaceted functions.

### 4.2. Functions of Cx43 Hemichannels in Normal and Tumor Cells

Beyond its established role in gap junctions, Cx43 constitutes non-coupling hemichannels on the plasma membrane, establishing a bridge between the intracellular milieu of isolated cells and the extracellular microenvironment [172]. The functional role of hemichannels in the tumor microenvironment remains largely unexplored [63], although this area is garnering attention due to its implication in cellular homeostasis, whereas its disruption is often a feature of malignant transformation [173,174,175].

Most studies on hemichannels were aimed at elucidating their function in physiological contexts, delving into their contribution to cellular processes such as calcium signaling, cell survival, proliferation, apoptosis, and differentiation [9,176]. This surge of interest in hemichannels may be linked to their role in maintaining cellular homeostasis and orchestrating intricate biological processes [177,178,179]. Notably, Cx43 hemichannels under normal resting conditions remain in closed conformation and have low open probability; however, their state is dynamically regulated by both intracellular and extracellular stimuli [180]. Hemichannel regulation is primarily dependent on ionic concentration, mainly calcium (Ca^2+^) gradients [181,182,183]. Additionally, hemichannel regulation can be brought about by membrane depolarization and metabolic changes, as well as mechanical membrane stress [176,184,185,186].

On opening, Cx43 hemichannels have the capacity to modulate autocrine and paracrine signaling by releasing molecules such as NAD+, glutamate, or ATP. This subsequently triggers the activation of the AKT/AMPK/mTOR signaling pathways, impacting on key cellular processes such as proliferation and survival [187,188]. Cx43 hemichannel-dependent modulation of Ca^2+^ concentrations and ATP extracellular release regulates the proliferation of multiple cell types [189,190]; this is often due to an increase of intracellular Ca^2+^ elicited by NAD+ and ATP signaling [63]. Additionally, increased intracellular Ca^2+^ concentrations have been found to inhibit GJIC, which is a common feature in malignant cells [176]. For instance, cadmium (Cd) exposure is associated with an increased risk of human prostate cancer, and a study conducted by Liu et al. demonstrated that human prostate epithelial cells RWPE-1 incubated with cadmium show increased Cx43 expression and intracellular Ca^2+^ levels, promoting cell migration [51]. Cyclic adenosine monophosphate (cAMP) is another small molecule acting as a secondary messenger involved in intracellular signal transduction processes [191]. Cx43 is furthermore implicated in the regulation of the cAMP/PKA signaling pathway; however, prior research has explored this function, mostly focusing on its role in facilitating the cell–cell exchange of cAMP molecules via GJIC [192,193]. Further investigations have suggested that cAMP pathway stimulation can elicit hemichannel functional switching, as well as endocrine and paracrine signaling [194,195]. At present, our understanding of the interplay between connexin hemichannels and cAMP/PKA signaling in cancer is limited; nevertheless, this line of inquiry holds promise.

In general, the function of connexin hemichannels in cancer is a topic of active investigation, though only a handful of studies have illustrated their role in the tumor microenvironment. They can facilitate the environmental uptake of crucial biomolecules like glucose and survival factors, as well as glutamate, supporting cell survival in nutrient-deficient conditions [196]. As previously mentioned, the study conducted by Khalil et al. proposes that purinergic ATP signaling mediated by Cx43 hemichannels controls the leader cells responsible for the collective migration of breast cancer cells [87]. Furthermore, another recent study suggests Cx43 hemichannels in osteocytes reduce the oxidative stress of tumor cells found in the osteocytic microenvironment (in a mouse model of bone metastasis), by facilitating the exchange of antioxidative molecules and supporting the activity of redox enzymes [197]. However, the regulation and functional role of hemichannels in tumor cells remain largely unexplored; in particular, since most reagents blocking gap junctions also block connexin hemichannels, the specific functional relevance of hemichannels is difficult to establish conclusively [63]. Nonetheless, their permeability to nucleotides and secondary messengers, along with the research findings described above, points to a key role of hemichannels in tumor growth and disease progression. Therefore, it is of significant value to delve deeper into this field, elucidating the engagement of hemichannels in the various steps of the metastatic cascade.

## 5. Therapeutic Potential of Targeting Cx43

Connexin-43, with its broad involvement in cancer, has a lucrative therapeutic potential; however, due to its complex functional role, specific interventional approaches need to be validated. Although recent data have suggested promising avenues, a deeper exploration is warranted to establish Cx43 targeting as a relevant treatment. Of note, studies have suggested Cx43 expression to be an independent prognostic marker to predict patient outcome. In this regard, a study carried out by Chasampalioti et al. proffered Cx43 expression to be an independent positive predictor of patient outcome and distant metastasis-free survival for breast cancer patients [86]. Other studies, however, have established Cx43 as either a positive or a negative biomarker for prognosis in different human cancers [41,90,109,198,199,200]. In fact, as previously mentioned, early-stage cancers have been reported to exhibit a loss or downregulation of Cx43 and GJIC activity; thus, restoring Cx43 expression and GJIC was undertaken as a treatment approach [14]. Notably, various chemotherapeutic agents have been reported to upregulate Cx43 expression and promote GJIC restoration [201]. For example, eicosapentaenoic acid (EPA) enhances Cx43 levels in melanoma cells, increasing chemosensitivity to 5-fluorouracil [202]. However, owing to the discussed multifaceted role of Cx43 in cancers, where in specific cases and more advanced stages it can facilitate malignant features and metastasis [2], restoring Cx43 expression may not represent a safe approach. Indeed, various studies aimed to target Cx43 to attenuate its pro-metastatic activity. For example, by means of Cx43 knockdown or treatment with gap junction channel inhibitors, such as tonabersat and meclofenamate, Chen et al. demonstrated inhibitory effects on brain metastases [56]. Notably, meclofenamate is in currently under clinical trial (NCT02429570) for treatment of recurrent or progressive brain metastases [175], and tonabersat shows promise in preclinical studies [203]. Moreover, since Cx43 hemichannels are also implicated in cancer progression (as elucidated in this article), a study conducted by León-Paravic et al. explores the use of Carbon Monoxide (CO) to inhibit Cx43 hemichannels for cancer cell treatment [204]. In general, taking into account the multifaceted activity of Cx43 in the tumor microenvironment, further pre-clinical studies in animal models are warranted.

A substantial limitation of many of these therapeutic strategies is their lack of specificity in targeting Cx43, implicating various side effects depending on the biological context. The use of peptides and monoclonal anti-Cx43 antibodies can offer greater specificity compared to conventional chemical agents, potentially reducing off-target effects, and the studies so far have yielded promising results [14,175]. However, the complex role of Cx43 in cancers, including gap junction-dependent and independent activities, presents significant therapeutic challenges. Therefore, more focused research is required to elucidate the mechanisms by which Cx43 influences tumor metastatic progression, and these insights will further aid in designing and optimizing targeted therapeutic approaches.

## 6. Conclusions

Intercellular communication through gap junctions is a crucial mechanism controlling cell functions in normal tissues, including cell growth, differentiation, and migration. The dysregulation of these channels, due to mutations or altered expression, is implicated in a wide range of diseases, notably in cancer. However, the role of connexin proteins, particularly Cx43, in cancer invasion and metastasis remains an active area of investigation. Available studies have underscored a range of functions of Cx43, transcending the canonical role in cell-to-cell communication to encompass intracellular signaling and the activity in hemichannels, gating the exchange of ions and small signaling molecules with the extracellular environment. These multifaceted functions of Cx43 are increasingly assigned a pivotal role in cancer, which this review endeavors to encapsulate. Highlighting these aspects can not only contribute to our understanding of the intricate mechanisms underlying cancer progression, but also serve as a foundation for future endeavors aimed at characterizing the specific role of Cx43 across the spectrum of human malignancies, ultimately fostering the progress of translational research for improved patient management and clinical outcomes.

## Figures and Tables

**Figure 1 cancers-16-04191-f001:**
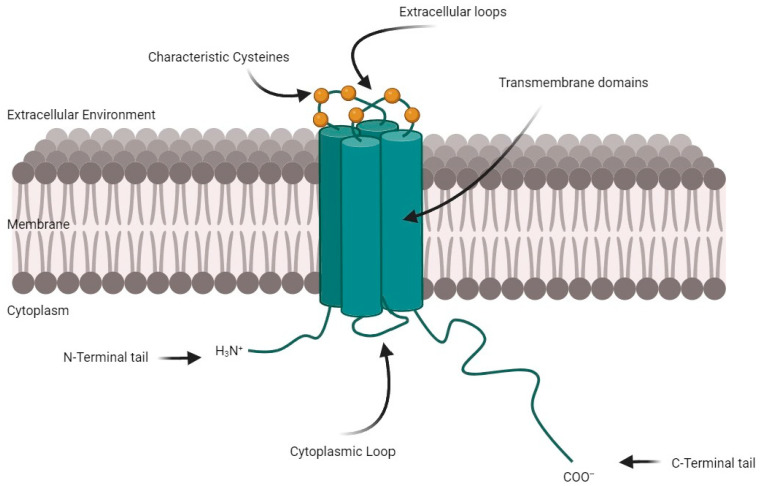
Cx43 protein structure. Cx43 comprises four transmembrane domains, with two extracellular loops, one cytoplasmic loop, and a cytoplasmic N-terminus and a long C-terminus tail. Illustrations generated using BioRender.com (accessed on 14 April 2024).

**Figure 2 cancers-16-04191-f002:**
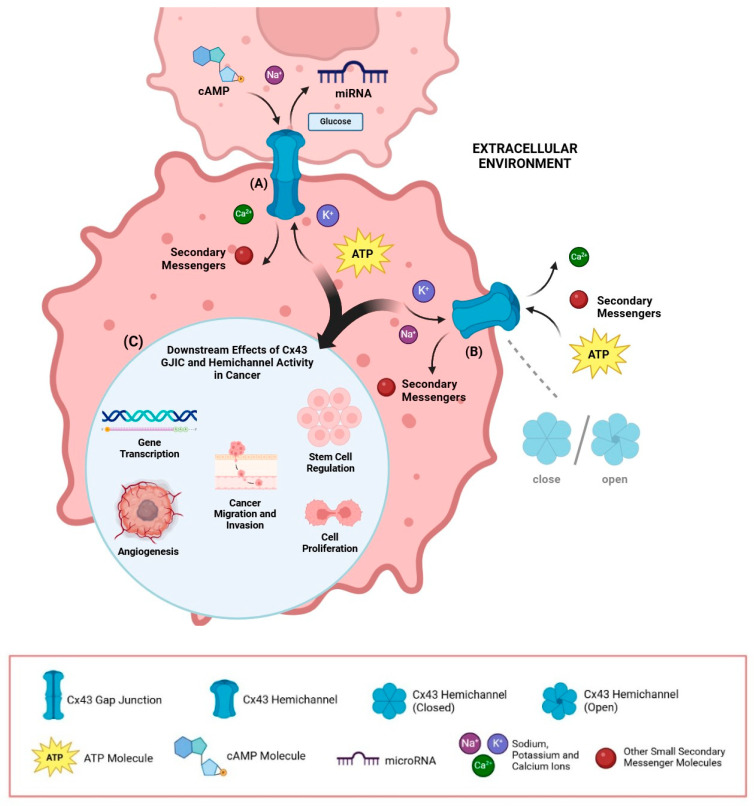
Schematic overview of Cx43’s activities in tumor cells. The figure summarizes the roles of Cx43 gap junctions and hemichannels in tumor progression. (**A**) Cx43-based gap junctions enable the direct intercellular exchange of small molecules, ions, and metabolites between tumor cells and the tumor microenvironment. (**B**) Cx43 hemichannels, typically closed, can open under specific stimuli (e.g., membrane depolarization, mechanical stress, and molecular regulation), mediating bidirectional communication between the intracellular and extracellular compartments, and influencing distant cells in the tumor microenvironment. (**C**) Together, these activities can regulate intracellular signaling cascades, enhancing metastatic behavior and promoting tumor progression. Illustrations generated using BioRender.com (accessed on 3 December 2024).

**Table 1 cancers-16-04191-t001:** Expression and Function of Cx43 in Different Cancers.

Cancer Type	Cx43 Expression	Functions in Cancer	Ref.
Breast Cancer	Expression levels vary, with loss/downregulation in primary tumors but upregulation in metastatic tumors.	Loss of Cx43 GJIC often found as an early event in malignancy. Also reported were mislocalization and increased expression in metastasis.	[40,62,81,84,86,87,88,89,90]
Colorectal Cancer	Significantly downregulated in tumor tissue.	Acts as a tumor suppressor, and furthermore, is involved in chemosensitivity.	[69,91,92,93,94,95,96]
Glioblastoma	Expression levels vary with stages of tumor progression.	Involved in regulation of cell growth, cancer metastasis, resistance to apoptosis, and chemoresistance.	[74,97,98,99,100,101,102,103,104,105,106,107,108]
Lung Cancer	Expression levels vary with stages of tumor progression.	Impacts cancer cell proliferation and migration, resistance to chemotherapy, and is often associated with poor prognosis.	[52,85,109,110,111,112,113,114,115,116,117,118,119]
Ovarian Cancer	Altered expression, often upregulated in metastatic stages.	Involved in cell migration, chemoresistance, and cancer cell aggressiveness.	[120,121,122,123,124,125,126,127]
Pancreatic Cancer	Altered expression, often downregulated.	Primarily involved as a tumor suppressor, promoting cancer cell apoptosis and inhibiting metastasis.	[128,129,130,131,132,133,134]

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
