# Peer review of "Connexin-43 in Cancer: Above and Beyond Gap Junctions!"

_cancers, 2024, doi:10.3390/cancers16244191_

Round 1

Reviewer 1 Report

Comments and Suggestions for Authors

Concerning manuscript cancers-3331338 entitled “Connexin-43 in Cancer: Above and Beyond Gap Junctions!”, this is an excellent review that provides a deep and complete understanding of what is known concerning both the general role and probable role of Connexin-43 in various cancers and how this might be targeted for therapy. It is very well written and presented and the figure does an excellent job of increasing our understanding of what is presented in the text. It will be of great help both for young scientists wanting to deepen their understanding of the biology of connexins and for estabilished scientists to consider how to possibily utilize this information in designing novel therapeutic strategies.

I also liked how they wrote on the possible conection with Tunneling nanotubes  and Cancer stem cells.

This review will be of great help both for young scientists wanting to deepen their understanding of the biology of connexins and for estabilished scientists to consider how to possibily utilize this information in designing novel therapeutic strategies.The English is correct and clear throughout the manuscript.

My complements on this well done job.

I believe that this excellent manuscript is ready for immediate publication.

Author Response

We are deeply grateful to this Reviewer for his/her appreciation of our work.

Reviewer 2 Report

Comments and Suggestions for Authors

This review summarized the activities of Connexin-43 (Cx43) in the context of non-canonical, gap junction-independent, functional involvement in cancer progression, focusing on how this information can promote our understanding of tumor metastases.

The paper is written clearly and concisely. The introduction clearly outlines the scope and objectives, the aims and research questions of the review paper clearly stated. The references are accurate and complete. Overall, this work is suitable for publication after addressing a few minor concerns.

Comments and suggestions:

To help readers understand the structure logically, the review is better to have clear subsections and headings. Especially in section 2 and 3.

For section 2. Connexin-43 (Cx43), the subsections can include the structure, function, and the role in cancer of Cx43.

For section 3. Connexin-43 in Cancer, the subsections can include the structure, function, and the role in cancer of Cx43. The subsections can be classified by cancer type or Cx43 function in Cancer as the author list in Table 1.

Overall, I recommend minor revisions to this review before its publication in Cancers.

Author Response

We are deeply grateful to this Reviewer for his/her appreciation of our work and constructive comments. We have better organized the content of the article with subsections and headings, as suggested.

Reviewer 3 Report

Comments and Suggestions for Authors

The paper explores the multifaceted roles of Connexin-43 (Cx43) in cancer biology. Cx43 is a well-studied gap junction protein involved in cell communication. It forms channels that allow the exchange of molecules between adjacent cells. Beyond its traditional role in gap junctions, Cx43 also forms hemichannels and has intracellular signaling functions through its C-terminal tail (Cx43-CT). Cx43 has both tumor-suppressive and pro-tumorigenic roles, depending on the context. It can facilitate cancer cell survival, migration, and invasion. Cx43 hemichannels release signaling molecules that can influence cancer cell behavior, including promoting metastasis. The Cx43-CT interacts with various proteins, influencing cell growth, differentiation, and migration. Cx43 is involved in the regulation of cancer stem cells, contributing to their aggressive behavior and chemoresistance. Understanding the diverse functions of Cx43 in cancer could lead to new therapeutic strategies targeting its non-canonical roles. The paper emphasizes the importance of Cx43 beyond its gap junctional communication, highlighting its potential as a target for innovative cancer therapies. Specific comments:

1.          The introduction provides a good background on Connexin-43 (Cx43) and its roles. It would be helpful to include more recent references to support the claims about the role of Cx43 in cancer.

2.          The discussion provides a good overview of the potential mechanisms by which Cx43 exerts its effects. However, it would benefit from a more detailed discussion on how Cx43 specifically influences cancer progression.

3.          The paper discusses the potential clinical implications of the findings. Including a comparison with existing treatments for cancer and discussing the translational potential of targeting Cx43 would add value.

4.          The conclusion summarizes the key findings well. However, it would be helpful to include a brief discussion on the limitations of the study and potential future research directions.

5.          The paper presents interesting findings on the multifaceted roles of Cx43 in cancer. Addressing the above comments would strengthen the manuscript and enhance its impact.

Author Response

Specific comments:

  1. The introduction provides a good background on Connexin-43 (Cx43) and its roles. It would be helpful to include more recent references to support the claims about the role of Cx43 in cancer.

Re: We are grateful to the Reviewer for his/her appreciation of our work. According to the suggestion, we further checked recently published literature concerning the involvement of Cx43 in cancer and discussed a dozen additional recent studies in the manuscript. Altogether, in our revised article, we refer to 38 papers published in the last four years, 13 of which were in 2023 and 6 in 2024. We believe this provides an accurate up-to-date picture of the knowledge in the field.     

  1. The discussion provides a good overview of the potential mechanisms by which Cx43 exerts its effects. However, it would benefit from a more detailed discussion on how Cx43 specifically influences cancer progression.

Re: Although Cx43-mediated mechanisms influencing cancer progression have not been fully elucidated, we have improved their description in our revised article. For example, we have better organized the contents by subsections and headings, addressing the various activities associated with Cx43 in cancer, including gap junction-related functions. Moreover, we included a new schematic Figure 2, summarizing a large part of the current knowledge in the field.

  1. The paper discusses the potential clinical implications of the findings. Including a comparison with existing treatments for cancer and discussing the translational potential of targeting Cx43 would add value.

Re: It is fair to say that Cx43 emerges from literature as an interesting potential target for therapy, although we lack actual evidence about the impact of its targeting in the clinical setting. Thus, in this article we have speculated about the potential clinical implications of fundamental studies tackling Cx43 activity in cancer cells. On the other hand, addressing the Reviewer’s concern, we do not believe that -at the current stage- Cx43 targeting may be considered as a comparable alternative to standard of care therapeutic approaches for human cancers. In the revised version, we have expanded the discussion about the translational relevance of tackling Cx43 in tumors in a new section focused on this topic, also addressing limitations to be kept in mind when designing targeting approaches.

  1. The conclusion summarizes the key findings well. However, it would be helpful to include a brief discussion on the limitations of the study and potential future research directions.

Re: In line with the previous point, following Reviewer’s suggestion, we have included a discussion of the potential limitations of Cx43-targeting approaches in the clinical setting. In particular, it should be taken into account the multifaceted activity of Cx43 in the tumor microenvironment, demanding further pre-clinical studies.

  1. The paper presents interesting findings on the multifaceted roles of Cx43 in cancer. Addressing the above comments would strengthen the manuscript and enhance its impact.

Re: We are grateful to the Reviewer for his/her constructive comments and suggestions, which have guided us to improve our work.